# COVID-19 Vaccination and Immunosuppressive Therapy in Immune-Mediated Inflammatory Diseases

**DOI:** 10.3390/vaccines11121813

**Published:** 2023-12-04

**Authors:** José M. Serra López-Matencio, Esther F. Vicente-Rabaneda, Estefanía Alañón, Ainhoa Aranguren Oyarzabal, Pedro Martínez Fleta, Santos Castañeda

**Affiliations:** 1Hospital Pharmacy Service, Hospital Universitario de La Princesa, IIS-Princesa, 28006 Madrid, Spain; estefania.alanon@salud.madrid.org (E.A.); ainhoa.aranguren@salud.madrid.org (A.A.O.); 2Rheumatology Service, Hospital Universitario de La Princesa, IIS-Princesa, 28006 Madrid, Spain; evicenter.hlpr@salud.madrid.org; 3Immunology Service, Hospital Universitario de La Princesa, IIS-Princesa, 28006 Madrid, Spain; pedro.martinez@salud.madrid.org

**Keywords:** COVID-19, vaccination, immunosuppressive therapy, immune-mediated inflammatory diseases (IMIDs), DMARDs, biological drugs, targeted synthetic DMARDs

## Abstract

The COVID-19 vaccination program has probably been the most complex and extensive project in history until now, which has been a challenge for all the people involved in the planning and management of this program. Patients with immune-mediated inflammatory diseases (IMIDs) on immunosuppressive therapy have required special attention, not only because of the particular haste in carrying out the process but also because of the uncertainty regarding their response to the vaccines. We now have strong scientific evidence that supports the hypothesis that immunosuppressive therapy inhibits the humoral response to vaccines against other infectious agents, such as influenza, pneumococcus and hepatitis B. This has led to the hypothesis that the same could happen with the COVID-19 vaccine. Several studies have therefore already been carried out in this area, suggesting that temporarily discontinuing the administration of methotrexate for 2 weeks post-vaccination could improve the vaccine response, and other studies with various immunosuppressive drugs are in the same line. However, the fact of withholding or interrupting immunosuppressive therapy when dealing with COVID-19 vaccination remains unclear. On this basis, our article tries to compile the information available on the effect of immunosuppressant agents on COVID-19 vaccine responses in patients with IMIDs and proposes an algorithm for the management of these patients.

## 1. Introduction

Severe acute respiratory syndrome coronavirus-2 (SARS-CoV-2) has recently caused a global pandemic with devastating consequences. Fortunately, after a period of uncertainty, several vaccines against SARS-CoV-2 have been developed. The purpose of these vaccines is to reduce the incidence of new infections and to minimize the morbidity, mortality and sequelae of the infection. By inducing an effective and long-lasting immune response, SARS-CoV-2 raised special concerns in physicians attending to patients with immune-mediated inflammatory diseases (IMIDs), such as rheumatoid arthritis (RA), psoriatic arthritis (PsA), psoriasis (Ps), systemic lupus erythematosus (SLE), inflammatory bowel disease (IBD) or multiple sclerosis (MS), as it might represent an important health challenge in this population. First, at the beginning of the SARS-CoV-2 pandemic, these patients were thought to be more susceptible to contracting the infection and developing worse clinical consequences; however, the importance of other factors, such as age or comorbidities, was identified later on. Additionally, the immune dysfunction related to this disease might be associated with a reduced immune response to COVID-19 vaccines, as well as to an increased risk of flares. Finally, the disease-modifying antirheumatic drugs (DMARDs) needed to control the activity of this disease might have a negative effect on the immune response (both cellular and humoral) to these vaccines, based on previous knowledge about other vaccines [1,2]. Furthermore, the initial lack of data on the efficacy and safety of the COVID-19 vaccine in immunosuppressed subjects due to their exclusion from clinical trials added uncertainty about whether to continue or temporarily withhold the DMARDs after a COVID-19 vaccination in an attempt to maximize its efficacy against SARS-CoV-2. In addition, this uncertainty was amplified by the novel mechanism of action of the mRNA vaccines and thus the scarce evidence of their long-term clinical activity [1]. In this context, the possibility that the immune response to the COVID-19 vaccine in patients with IMIDs could be altered and that DMARDs, such as methotrexate, among others, could exert a negative effect on its efficacy was more than plausible; this hypothesis was supported by several studies [3].

The purpose of this review is to analyze and discuss the current evidence on the COVID-19 vaccine immune response, both humoral and cellular, in patients with IMIDs receiving immunosuppressive medication, with a special emphasis on RA. Additionally, we searched for evidence of the efficacy of the vaccine in terms of reducing the incidence of severe COVID-19, hospitalizations and mortality.

## 2. Methods

A literature review from the PubMed database was performed by the authors between 1 August 2022 and 29 August 2023. The following terms were combined in the search strategy: COVID-19 vaccine, immunogenicity, efficacy, IMIDs, rheumatoid arthritis, immunosuppressants, glucocorticoids, DMARDs and their specific groups (methotrexate, biologics, tumor necrosis factorα inhibitors, anti-CD20, Janus kinase inhibitors, abatacept, interleukin-6 receptor inhibitors). The studies had to meet the following inclusion criteria to be selected: (a) be original articles (such as clinical trials, observational studies and case series) and (b) evaluate the immunogenicity and efficacy of the COVID-19 vaccine in patients with IMIDs being treated with DMARDs. Reviews, systematic reviews or recommendations lacking original data as well as case reports or studies that focused exclusively on safety were excluded. After a review of the abstract, articles that did not meet the inclusion criteria were excluded. Afterward, the complete reading of the works was carried out, with the selection of the articles finally being included in this narrative review. This review study did not require ethical approval.

## 3. Results

### 3.1. Disease-Modifying Antirheumatic Drugs (DMARDs)

There is currently little information in the literature regarding the effect of conventional synthetic (cs)DMARDs on patients’ immune responses to the COVID-19 vaccine. Previous evidence with other vaccines in this regard suggests the existence of a different impact is dependent on the DMARD used. Azathioprine (antimetabolite), sulfasalazine, hydroxychloroquine or leflunomide seem to reduce the antibody titers generated upon the administration of the influenza vaccine, but without affecting the seroprotection conferred by the vaccine [4,5]. Mycophenolate mofetil, an inhibitor of DNA synthesis and thus of the proliferation of T and B-lymphocytes, was shown to be associated with a reduction in antibody titers generated by the influenza vaccine in a trial, although below the limit necessary to have a significant impact on the immune response [4].

MTX—which inhibits dihydrofolate reductase, prevents the reduction of dihydrobiopterin (BH2) to tetrahydrobiopterin (BH4) and increases the sensitivity of T cells to apoptosis (diminishing immune responses)—is the drug that has been most extensively studied in the literature and it has been associated with a reduction in the humoral response generated by influenza and pneumococcal vaccines [6]. In fact, MTX is used in clinical practice to reduce the formation of human anti-chimeric antibodies (HACA) [7]. The randomized studies that have investigated the effect of temporarily withdrawing MTX after the influenza vaccination are of special interest. They suggest that the critical period in the generation of the humoral response is 2 weeks after vaccination. Of the strategies investigated, the interruption of MTX 2 weeks after vaccination is the one that obtains the best results, enhancing the humoral response to the vaccine, without having a clinically relevant impact on the activity of the disease; longer treatment breaks (4 weeks) do not seem to increase this benefit and are associated with a greater risk of a flare [8].

Based on these data, some researchers have investigated the impact of MTX on the COVID-19 vaccine, finding similar results that support the possibility of establishing strategies to enhance the response to the COVID-19 vaccine in patients with IMIDs who receive MTX and open the door to investigate the impact of analogous strategies with other csDMARDs and other vaccines. One randomized clinical trial including 138 patients with RA showed that a 2-week MTX withdrawal after each dose of the COVID-19 vaccine improves the anti-SARS-CoV-2 IgG response when compared to maintaining MTX unchanged [9]. In another open label, a superiority clinical trial including 254 patients, which were randomized 1:1 to continue MTX or discontinue it for two weeks at the time of vaccination, with RA and psoriasis on MTX treatment found an enhanced boosting of antibody responses in the second group [10]. Two additional trials assessing the effect of withholding methotrexate for 2 weeks after the first and second doses (MIVAC I) or after the second dose (MIVAC II) of the COVID-19 vaccine in patients with RA or psoriatic arthritis (PsA) resulted in a higher humoral response measured 4 weeks after vaccination in the withdrawal arm than in the group that maintained MTX unchanged [11].

Observational studies have also studied the impact of MTX on the immune response to the COVID-19 vaccine and identified additional factors involved. One retrospective study concluded that MTX reduces the humoral response after the COVID-19 vaccination in an age-dependent manner [12]. In the same line, Haberman et al. showed that MTX hindered the immune response in adults with autoimmune diseases [13]. Recently, the same group has carried out a study that indicates that this reduction in the humoral response can be avoided with a week of rest from MTX [14], with similar findings documented by other authors in an observational study including 136 patients with RA [15]. Furthermore, Frommert et al. have reported additional factors to MTX with a negative impact on the humoral immune response to the COVID-19 vaccine, such as the type of vaccine, the dosage interval or age [16].

Regarding the cellular response, data are scarcer than for humoral immunogenicity. Schmiedeberg et al. found a similar T cell response in a treated RA population and healthy controls after the COVID-19 vaccine, although it declined earlier in patients with RA [17]. The preservation of the cellular response reaffirms the positive role of the COVID-19 vaccine even in patients in whom the humoral response may foreseeably be more affected due to the disease or its treatments, although it would be of great interest to know if temporary DMARD rest strategies also enhance cellular immunity in these patients.

A summary of the main studies on COVID-19 vaccine immunogenicity in patients with IMIDs undergoing chronic DMARDs therapy is shown in Table 1 [10,13,18,19,20,21,22,23,24,25,26,27,28,29].

### 3.2. Glucocorticoids

Glucocorticoids have several immunomodulatory effects, such as the inhibition of the synthesis of pro-inflammatory cytokines, reduction of leucocyte trafficking and induction of the apoptosis of T-lymphocytes. Concerning the modification of the immune response to vaccines in patients undergoing chronic treatment with GCs, it should be remembered that it is dose-dependent. In fact, doses of prednisolone ≥ 10 mg/d have been associated with a decrease in the patient’s humoral response to the 23-valent pneumococcal polysaccharide vaccine and a higher rate of infections [30]. It is also noteworthy that prednisolone doses higher than 10 mg/d have been associated with worse disease progression in patients with COVID-19, indicating that cellular immunity could also be affected [31], although the use of GCs can improve the patient’s prognosis, especially in severe pneumonia caused by COVID-19 [32]. Interestingly, several mechanisms underlying the beneficial effects of dexamethasone during severe COVID-19 have been postulated: it affects circulating neutrophils, alters IFN active neutrophils, downregulates interferon-stimulated genes and activates IL-1R2+ neutrophils in severe COVID-19 patients [33].

Humoral and cellular immune responses to COVID-19 in patients undergoing GC therapy show contradictory results. A recent study on patients with SLE concludes that the medium-term response of these patients to the SARS-CoV-2 vaccination may be compromised by GC use and by other prescribed treatments aimed to control the severity of the disease, such as rituximab, identifying the GC therapy as the factor most associated with declining levels of neutralizing antibodies induced by the vaccine [34]. On the contrary, another study concludes that COVID-19 vaccines are immunogenic in patients receiving immunosuppression, when assessed by a combination of serology and cell-based assays, despite the response being impaired compared to healthy individuals [35].

### 3.3. Tumor Necrosis Factor-α Inhibitors

Tumor necrosis factor alpha (TNF-α) is a pro-inflammatory cytokine that recruits neutrophils and monocytes to the inflammation areas and activates intracellular signaling in several cells of the immune system. There are numerous studies on the immune response to different types of vaccines when administered concomitantly with TNF inhibitors. Most of these studies did not find a significant decrease in the immune response to pneumococcal, influenza and varicella vaccines [3,4,36,37]. Regarding the impact of TNF inhibitors on the immune response to the COVID-19 vaccine, the quality of the evidence is low and does not allow for firm conclusions to be drawn since some authors have described diminished serological and cellular responses in these patients, and others have reported that immunogenicity is not hampered.

Table 2 shows a summary of the main studies on the immune response to the COVID-19 vaccine in patients with biological treatments [18,19,20,21,22,23,25,26,27,28,29,38,39,40,41,42,43,44,45,46,47].

### 3.4. ANTI-CD20

B-cells are an essential component of the adaptive immune system. In fact, CD-20 depletion produces a clear decrease in the immune response to certain vaccines, such as influenza or pneumococcal [4,24], as it is to be expected given the key role of B cells in humoral immunity.

Evidence from the literature indicates that RTX is associated with a worse evolution of COVID-19 and an increased risk of hospitalization [48,49]. Nevertheless, it is worth noting that RTX-treated patients may have additional risk factors contributing to these worse outcomes. In this context, the data on the cellular response to the COVID-19 vaccine in patients with IMIDs treated with rituximab are especially relevant, given the foreseeable negative effect on the humoral response due to its mechanism of action. As expected, a negative association between seroconversion rates to the COVID-19 vaccine and rituximab has been confirmed, identifying the interval between the last RTX infusion and the first vaccination, the number of peripheral B-cells and the immunoglobulin quantity among the related factors [50,51], while the cellular response seems to be preserved [52]. Intervals of 6 to 9 months between the last rituximab administration and COVID-19 vaccination appear to improve the humoral response, although pharmacokinetic studies suggest that the presence of B cells and/or rituximab in the blood predict seroconversion better than time since last infusion [50,51,53]. The evidence points to the special need to individualize the COVID-19 vaccination strategy in patients with IMIDs being treated with rituximab, although it does not seem advisable to delay primary immunization in cases in which it is not advisable to delay treatment due to the activity of the disease [54].

### 3.5. Janus Kinase Inhibitors

The Janus kinase (JAK)/signal transduction and activators of transcription (STAT) pathway is responsible for signal transductions triggered by a number of cytokines and growth factors. Thus, theoretically, this family of drugs may be problematic in the case of the COVID-19 infection, given its mechanism of action. Previous evidence shows that the effect of a pneumococcal conjugate and a tetanus toxoid vaccine in baricitinib-treated patients results in lower and less durable immune responses [55]. However, there is no clear evidence in the case of tofacitinib.

Data about the influence of JAK inhibitors on COVID-19 vaccine are scarce and focused on the humoral response. A retrospective French registry found an overall serological response rate of 88%, which was negatively affected by older age and a relatively higher rate of non-responders among upadacitinib users in comparison with baricitinib or tofacitinib [56]. However, other studies have found a lower humoral response in patients undergoing treatment with JAK inhibitors than in patients treated with csDMARDs or healthy controls, with a deeper reduction in neutralizing antibody titers when administered in combination with methotrexate, suggesting the potential beneficial role of a temporary discontinuation of the JAK inhibitors after vaccination [57,58].

### 3.6. Interleukin-6 Receptor Inhibitors

Interleukin-6 (IL-6) is a very important pro-inflammatory cytokine, and is involved in the acute phase response and in the differentiation and function of B and T cells. The response to pneumoccocal and influenza vaccines in anti-interleukin-6 receptor (anti-IL-6R)-treated patients, with or without MTX, has been assessed and showed no differences with the control group when anti-IL-6 receptor inhibitors were used in monotherapy, while presenting an impairment of the immune responses if combined with MTX [59].

Regarding the COVID-19 vaccine, SARS-CoV-2-specific IgG1 titers seem to depend on disease severity and not on tocilizumab treatment [60]. If we extrapolate these results to the vaccination response, we could hypothesize that humoral immunogenicity may not be affected by IL-6 receptor inhibitors, though this would be a mere assumption. In this line, a recent retrospective cross-sectional multicenter study in neuromyelitis optica spectrum disorder (NMOSD)/myelin oligodendrocyte glycoprotein antibody-associated disease (MOGAD) patients treated with anti-IL-6 receptor therapy found a 100% seroconversion rate after a COVID-19 vaccination, despite titers being lower than in healthy controls, although comparable to csDMARDs and higher than B-cell depleting agents [61].

### 3.7. Abatacept

Abatacept is a fusion protein of the cytotoxic T lymphocyte-associated antigen-4 that selectively modulates the CD80/CD86:CD28 costimulatory signal required for full T-cell activation. Data existing so far regarding the efficacy of the vaccination in patients undergoing abatacept therapy have shown an impaired humoral response to both the influenza [62] and pneumococcal vaccines [63]. However, these results should be considered with caution due to the small sample size and the absence of a control group.

Patients with IMIDs treated with abatacept have shown to display a reduced humoral and cellular response to the COVID-19 vaccine, in contrast with rituximab and belimumab, which only affect the seroconversion and reduce the titers of neutralizing antibodies regarding the T cell response, or mycophenolate and azathioprine which impair antibody production, while leflunomide and anti-cytokine therapies seem to minimally hamper immunogenicity [64,65,66,67]. Notably, the administration of second or third doses of the COVID-19 vaccine to non-responders induced a more robust antibody response and an increased T cell response [64,65,67].

## 4. Discussion

Concerns about how to manage vaccination in patients with IMIDs have long existed [68]. Regarding the COVID-19 vaccine, preliminary progress has been made with the development of guidelines by several scientific societies [69,70,71,72]. However, a universally accepted consensus still needs to be reached, requiring a standardization of the proposed protocols (Table 3).

Current evidence indicates that seroconversion rates after vaccination against COVID-19 are lower in patients with IMIDs on immunosuppressive therapy than in healthy subjects despite the administration of a second dose. Thus, COVID-19 booster vaccines after a primary series have been used, and some evidence on their efficacy elicited seroconversion in non-responders without major adverse events to date [73].

Most of the patients with IMIDs are undergoing treatment with DMARDs, even in combination, and we have previously reviewed that some DMARDs have been associated with a reduced response to the COVID-19 vaccine, especially when used in combination. Accordingly, the type of DMARD received has to be taken into consideration because a different immune response profile to the COVID-19 vaccine has been found for each family of drugs. Anti-CD20 drugs have the lowest seroconversion rate, followed by abatacept and JAK inhibitors, and their impact seems to be deepened by the combination of methotrexate [74].

Therefore, the most coherent approach seems to act according to the type of DMARD and the patient’s situation at the time of vaccination to maintain a balance between enhancing the protection against the SARS-CoV-2 infection and not compromising the evolution/activity of the disease. Considering MTX, a temporal withdrawal of 2 weeks after the administration of the vaccine could help improve the immune system response to the COVID-19 vaccine. This strategy could be considered safe in patients in remission or with a low disease activity, as the risk of flares may be low. However, if the patient is active, the approach might be completely different in the following way: priority could be placed on the control of the disease and, therefore, DMARDs would maintain unchanged or a reduction in the interval of methotrexate interruption to one week would be applied, according to the clinical situation. In fact, we cannot forget that the mere fact of having previously suffered from the COVID-19 infection can worsen the course of several autoimmune diseases [75]. Therefore, in certain patients, it would not be indicated to interrupt the administration of the immunomodulatory agent. Nevertheless, patients infected by SARS-CoV-2 previously to being administer a vaccination have been described to develop higher response rates and titers of induced antibodies, making it less necessary to temporarily interrupt their DMARDs.

The GC dose should be as low as possible to improve the immune response to the vaccine, ideally <10 mg/d; however, the activity and course of the disease must be considered.

Evidence about other DMARDs is scarcer but points towards the same direction, finding a better response to the vaccine after withholding the administration of JAK inhibitors for only one week post-vaccination, and the same could be applied to abatacept given the evidence previously mentioned in the corresponding section. Anti-cytokine drugs, such asIL-6 receptor inhibitors and TNF-α inhibitors, require more in-depth research to reach a consensus about the best timing of these therapies when administering the COVID-19 vaccine, although preliminary data suggest they minimally hamper vaccine-induced immunogenicity.

However, there are clinical efficacy parameters that we can consider to be of greater weight than the immunogenicity response to the vaccine, such as the efficacy against symptomatic COVID (especially severe COVID), ICU, hospital and emergency admissions and mortality. We have less information in this regard, although reviews are beginning to be published showing substantial reductions in all of them, in the following order: 50%–87%, 55%–100% and 82%–87%, respectively [76,77,78,79]. In patients with IMIDs, a 0% incidence of post-vaccination COVID-19 has been described in a study that included 158 cases [80]; however, studies with larger sample sizes and longer follow-ups are needed to draw conclusions about the true impact of DMARDs and the temporary rest strategies from these treatments on relevant clinical outcomes, such as hospitalizations and mortality.

### Limitations

There is insufficient evidence to generate robust recommendations for most of the drugs involved due to the scarcity of the data and the low quality of the evidence that comes from uncontrolled retrospective observational studies, with the exception of the randomized studies about methotrexate. Additionally, the immune response to COVID-19 vaccines is affected by many factors, not only the immunosuppressive therapy, such as the type of immune disease, demographic factors, comorbidities, other therapies or the event of a prior COVID-19 infection (among others), making the interpretation of the results more complex. Furthermore, many patients receive a combination of therapies that include glucocorticoids and more than one immunosuppressive drug, adding extra difficulties to the analysis of the results. The activity of the rheumatic disease at vaccination is another key factor that can have an impact on the immune response to vaccines and was not clearly described in some of the studies. Given the limitation of the data, caution and an individualized approach to immunosuppressive therapy management during COVID-19 vaccination should be advised.

Figure 1 shows the main factors that may influence the management of these patients when designing a COVID-19 vaccination strategy based on the currently available knowledge [81].

## 5. Summary

Vaccination is one of the most important strategies that have been used to mitigate the damage of the global COVID-19 pandemic. In fact, it has been decisive in overcoming the worst pandemic that humanity has suffered in recent history. Moreover, all this has been thanks to, on the one hand, its great effectiveness and, on the other, to the great coordination of the parties involved. Patients with IMIDs may be especially prone to a more severe COVID-19 infection due to their disease and therapies and the associated comorbidities. Therefore, the COVID-19 vaccination should be prioritized in these patients. It has been documented that the use of DMARDs, especially MTX and drugs that suppress the CD-20-positive lymphocyte population, may decrease the humoral and cellular response rate of the COVID-19 vaccine. The potential efficacy of the strategies that temporarily withdraw these DMARDs for a short period of time after COVID-19 vaccination deserves to be highlighted in patients with IMIDs under control, as it seems not be associated with relevant safety issues. Therefore, vaccination timing and DMARD administration adjustments should be planned in order to optimize the immune response to the COVID-19 vaccine in patients with IMIDs, whenever possible.

The potential clinical efficacy of strategies based on the adjustment of the administration of DMARDs during vaccination, the relevance of the cellular response in patients with IMIDs, the duration of the humoral and cellular responses or the need of periodic boosters of the COVID-19 vaccine in these patients are issues that still need to be clarified. Therefore, it is important to continue working on the generation of evidence on whether or not to delay DMARD administration at the time of vaccination. Although, at present, the COVID-19 infection seems to have been overcome, we are never exempt from the appearance of another similar pandemic wave. Thus, it is essential to be prepared by having clear, efficient and standardized vaccination strategies.

Overall, any vaccination needs to be individualized to each patient, weighing the benefits of transiently stopping DMARDs against the risks of potential flares of the disease. The evidence available in the literature is in favor of this strategy as it has been associated with enhanced humoral and cellular responses to the COVID-19 vaccine, with a scarce incidence of flares, which were mostly mild in intensity. However, more research is needed on the clinical relevance of these findings in terms of decreasing the incidence and severity of subsequent COVID infections, although preliminary data suggest a trend towards a beneficial effect.

## Figures and Tables

**Figure 1 vaccines-11-01813-f001:**
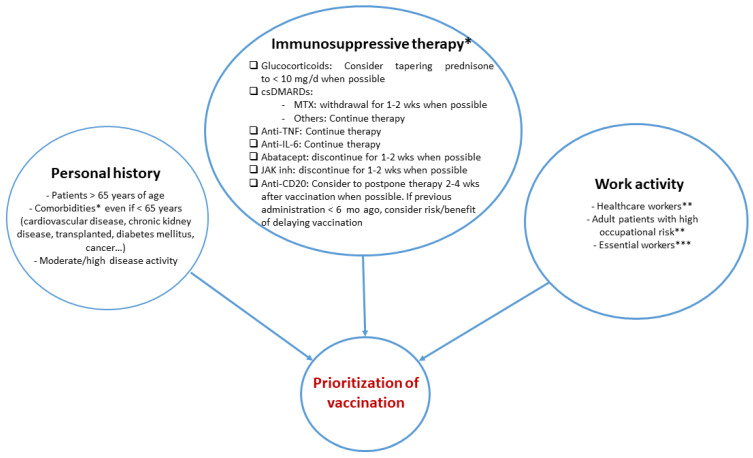
Main factors that may influence the management of immunosuppressed a patient when designing a vaccination strategy for COVID-19. Footnotes: (cs)DMARDs: (conventional synthetic) disease-modifying antirheumatic drugs; IL: interleukin; JAK: Janus kinase; mo: month; MTX: methotrexate; TNF: tumor necrosis factor; wks: weeks. * Proposal of the authors considering the available bibliography. ** According to the ECDC (European Centre for Disease Prevention and Control) Technical Report: https://www.ecdc.europa.eu/sites/default/files/documents/COVID-19-vaccination-and-prioritisation-strategies.pdf (accessed on 10 October 2023). *** According to the Spanish government: https://www.mites.gob.es/ficheros/ministerio/contacto_ministerio/lista_actividades_esenciales.pdf (accessed on 10 October 2023) [82].

**Table 1 vaccines-11-01813-t001:** Main studies on the COVID-19 vaccine immunogenicity in patients with IMIDs receiving DMARDs.

First Author and Reference Number	Pts (N)	Age (Median)	Disease	DMARDs
Abhishek A et al. [10]	340	59	RA, PsA	MTX
Ammitzbøll C et al. [18]	134	70	SLE, RA	MTX
Boyarsky BJ et al. [19]	123	50	IJDs, SLE, SS, myositis, vasculitis	AZA, HCQ, MMF, SSZ, TAC, MTX, leflunomide
Braun-Moscovici Y et al. [20]	264	50	IJDs, CTDs, vasculitis	MTX, MMF, GC
Bugatti S et al. [21]	140	55.7	RA, PsA, SpA	MTX, SSZ, leflunomide, cyclosporine A
Furer V et al. [22]	686	59	RA, PsA, SpA, SLE, vasculitis, LVV, AAV	GC, MTX, HCQ, MMF, leflunomide
Haberman RH et al. [13]	77	36	Lymphoma	MTX
Kappelman MD et al. [23]	317	50.9	IBD	GC, 5-ASA, SSZ, thiopurines
Mrak D et al. [24]	74	61.7	IgG4-related, CTDs, RA, vasculitis	MTX, MMF, HCQ, AZA, SSZ, GC leflunomide
Deepak P et al. [25]	133	45.5	IBD, IJDs, RA, SpA, SLE, SS, psoriasis, PsA	AZA, MMF, MTX, GC, leflunomide
Ruddy JA et al. [26]	404	384	Myositis	MMF, GC
Shenoy P et al. [27]	449	52	RA, IJDs, SpA, SLE, PR, scleroderma vasculitis, myositis	MTX, SSZ, leflunomide, HCQ, MMF, GC, TAC, AZA
Simon D et al. [28]	84	53.1	IBD, RA, SpA, psoriasis	5-ASA, HCQ, MTX, GC
Veenstra J et al. [29]	74	55.9	IBD, RA, SLE, psoriasis, PsA	HCQ, AZA, MMF, GC

Abbreviations: AAV: anti-neutrophil cytoplasmic autoantibody (ANCA)-associated vasculitis; 5-ASA: 5 amino salicylates; AZA: azathioprine; CTDs: connective tissue diseases; GC: gucocorticoids; HCQ: hydroxychloroquine; IBD: inflammatory bowel disease; IMIDs: immune-mediated inflammatory diseases; IJDs: inflammatory joint diseases; LVV: large vessel vasculitis; MMF: mycophenolate mofetil; MTX: methotrexate; PR: palindromic rheumatism; PsA: psoriatic arthritis; Pts: patients; RA: rheumatoid arthritis; SSZ: sulfasalazine; SLE: systemic lupus erythematosus; SpA: spondyloarthritis; SS: Sjogren’s syndrome; TAC: tacrolimus.

**Table 2 vaccines-11-01813-t002:** Main studies on the management of COVID-19 vaccine in patients receiving biological DMARDs.

First Author and Reference Number	Pts (N)	Age (Median)	Disease	bDMARDs/Targeted Therapies
Al-Janabi A et al. [39]	120	53	Psoriasis, PsA, RA, SLE, Cröhn’s	Abatacept, adalimumab, brodalumab, certolizumab, etanercept, guselkumab, ixekizumab, risankizumab, secukinumab, tildrakizumab, ustekinumab
Ammitzbøll C et al. [18]	134	70	SLE, RA	Infliximab, adalimumab, JAKi, rituximab, tocilizumab, abatacept, belimumab
Boyarsky BJ et al. [19]	123	50	IJDs, SLE, SS, myositis, vasculitis	Abatacept, belimumab, rituximab, infliximab, adalimumab, tofacitinib
Braun-Moscovici Y et al. [20]	264	50	IJDs, CTDs, vasculitis	Rituximab, belimumab, infliximab, adalimumab, abatacept, JAKi
Bugatti S et al. [21]	140	56	RA, PsA, SpA	Infliximab, adalimumab, tocilizumab, guselkumab, secukinumab, JAKi, CTLA4-Ig
Dailey J et al. [40]	33	-	IBD	Vedolizumab, infliximab
Furer V et al. [22]	686	59	RA, PsA, SpA, SLE, IIM, vasculitis, LVV, AAV vasculitis	Infliximab, adalimumab, tocilizumab, rituximab, guselkumab, abatacept, JAKi, belimumab
Geisen UM et al. [41]	26	51	PsA, RA, MCTD, SpA, SLE, IBD, psoriasis, myositis, vasculitis, sarcoidosis	Infliximab, adalimumab, golimumab, certolizumab, etanercept, tocilizumab, vedolizumab, secukinumab, ustekinumab, ixekizumab, belimumab
Kappelman MD et al. [23]	317	51	IBD	Vedolizumab, ustekinumab
Kennedy NA et al. [42]	1293	44	IBD	Infliximab, vedolizumab
Mahil SK et al. [38]	84	43		Infliximab, adalimumab, guselkumab
Deepak P et al. [25]	133	46	IBD, IJDs, RA, SpA, SLE, SS, psoriasis, PsA	Infliximab, adalimumab, golimumab, abatacept, vedolizumab, ustekinumab, tofacitinib, rituximab, tocilizumab
Rubbert-Roth A et al. [43]	51	64	RA	Abatacept, JAKi
Ruddy JA et al. [26]	404	44	Myositis	Infliximab, adalimumab, rituximab
Seyahi E et al. [44]	104	42	RA, SpA/IBD, vasculitis, CTDs	Rituximab and various biological agents
Shenoy P et al. [27]	449	52	RA, PR, IJDs, SpA, SLE, vasculitis, scleroderma, myositis	Tofacitinib, apremilast, rituximab, adalimumab
Simon D et al. [28]	84	53	IBD, RA, SpA, psoriasis	Infliximab, adalimumab, guselkumab, secukinumab, JAKi, tocilizumab
Spiera R et al. [45]	89	61	RA, SLE, SS, PsA, vasculitis, myositis, MCTD, scleroderma	Adalimumab, etanercept, abatacept, secukinumab, JAKi, rituximab, tocilizumab, belimumab, anakinra
Veenstra J et al. [29]	74	56	IBD, RA, SLE, psoriasis, PsA	Infliximab, tofacitinib, ixekizumab
Westhoff TH et al. [46]	9	64	Rituximab-treated pts	Rituximab
Wong SY et al. [47]	26	-	IBD	Infliximab, adalimumab, vedolizumab, ustekinumab

Abbreviations: AAV: anti-neutrophil cytoplasmic autoantibody (ANCA)-associated vasculitis; bDMARDs: biological disease-modifying antirheumatic drugs; CTLA4-Ig: cytotoxic T lymphocyte-associated protein-4 immunoglobulin; IBD: inflammatory bowel disease; IJDs: inflammatory joint diseases; MCTD: mixed connective tissue disease; JAKi: Janus kinase inhibitors; LVV: large vessel vasculitis; PR: palindromic rheumatism; PsA: psoriatic arthritis; Pts: patients; RA: rheumatoid arthritis; SLE: systemic lupus erythematosus; SpA: spondyloarthritis; SS: Sjogren’s syndrome.

**Table 3 vaccines-11-01813-t003:** Main COVID-19 vaccination guidelines in rheumatic patients with inflammatory immune-mediated and joint diseases.

Year	Author/Institution [Reference Number]	Title
2021	European League Against Rheumatism (EULAR) [69]	EULAR recommendations for the management and vaccination of people with rheumatic and musculoskeletal diseases in the context of SARS-CoV-2: the November 2021 update
2021	Asia Pacific League of Associations for Rheumatology (APLAR) [70]	Updated APLAR consensus statements on care for patients with rheumatic diseases during the COVID-19 pandemic
2023	American College of Rheumatology (ACR) [71]	American College of Rheumatology Guidance for COVID-19 Vaccination in Patients With Rheumatic and Musculoskeletal Diseases: Version 5
2023	World Health Organization (WHO) [72]	Updated WHO Guidance for Prioritizing COVID-19 Vaccines

## Data Availability

Not applicable.

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
