# Peer review of "COVID-19 Vaccination and Immunosuppressive Therapy in Immune-Mediated Inflammatory Diseases"

_vaccines, 2023, doi:10.3390/vaccines11121813_

Round 1
Reviewer 1 Report
Comments and Suggestions for Authors
Authors reviewed the relationship between COVID-19 vaccination effects and immunosuppresive, drugs during the pandemic. Although the manuscript contains several important data, there are some major issues:
- Authors should summarize several shared characteristics of immunosuppressive drugs and concise the whole manuscript as a mini review. Several information could be demonstrated within the tables.
- Authors should emphasize primarily the efficacy of vaccines in the pandemic, but not solely on immune responses. In this regard, especially the conclusion as well as the corresponding figure should be revised.

Some minor editing is needed
Author Response
Authors reviewed the relationship between COVID-19 vaccination effects and immunosuppresive, drugs during the pandemic. Although the manuscript contains several important data, there are some major issues:
Answer: We would like to thank the reviewer for all the positive appraisal of our study and his/her comments. We very much appreciate the careful revision of our manuscript, which has allowed us to improve it and add answers to all the questions raised. Please, find below all the point-by-point replies to the queries requested.
- Authors should summarize several shared characteristics of immunosuppressive drugs and concise the whole manuscript as a mini review. Several information could be demonstrated within the tables.
Answer: We thank the reviewer for his/her suggestion. Therefore, we have proceeded to reduce the manuscript as much as possible to make it clearer and more concise (Data erased: lines 32-47…pages 1-2, and lines 349-352, page 10, of the revised manuscript).
- Authors should emphasize primarily the efficacy of vaccines in the pandemic, but not solely on immune responses. In this regard, especially the conclusion as well as the corresponding figure should be revised.
Answer: We thank the reviewer for his/her suggestion. Therefore, we have added the following paragraph to the text (lines 397-400 page 11, of the revised manuscript). “In fact, it has been decisive in overcoming the worst pandemic that humanity has suffered in recent history. And all this has been thanks, on the one hand, to its great effectiveness and, on the other, to the great coordination of the parties involved”.
Reviewer 2 Report
Comments and Suggestions for Authors
The manuscript “COVID-19 vaccination and immunosuppressive therapy in immune-mediated inflammatory diseases” by José M. Serra López-Matencio et al. summarized the current evidence on the COVID-19 vaccine response of patients with IMIDs receiving immunosuppressive medication. However, there are several concerns with this manuscript:
Major:
1. Does all the immunosuppressive medications affect COVID-19 vaccination? How did it work? Do authors have any idea or any publications about the specific mechanisms of suppressing the immune system?
2. The critical period in the generation of the humoral response is 2 weeks after vaccination. The interruption of MTX 2 weeks after vaccination is the one that obtains the best result, enhancing the humoral response to the vaccine. Is this good for all Rheumatoid Arthritis patients no matter how severe the symptom it is? How to evaluate?
Author Response
The manuscript “COVID-19 vaccination and immunosuppressive therapy in immune-mediated inflammatory diseases” by José M. Serra López-Matencio et al. summarized the current evidence on the COVID-19 vaccine response of patients with IMIDs receiving immunosuppressive medication. However, there are several concerns with this manuscript:
Major:
- Does all the immunosuppressive medications affect COVID-19 vaccination?
Answer: We thank the reviewer for his/her appreciation In theory, all immunosuppressive agents should affect in one way or another the response to vaccination, that is why in the article we review all the evidence so far to see if this is reflected in practice.
How did it work? Do authors have any idea or any publications about the specific mechanisms of suppressing the immune system?
Answer: We agree with the reviewer's comment and therefore we have added the immunosuppressive mechanism of action of each family of drugs at the beginning of each section as follow:
“some antimetabolites as” (line 120, page 3, of the revised manuscript).
“(an inhibitor of DNA synthesis and thus of the proliferation of T and B-lymphocytes)” (line 124, page 3, of the revised manuscript).
“that inhibits dihydrofolate reductase, preventing the reduction of dihydrobiopterin (BH2) to tetrahydrobiopterin (BH4), increasing sensitivity of T cells to apoptosis (diminishing immune responses)” (lines 127-129, page 3, of the revised manuscript).
“Glucocorticoids have several immunomodulatory effects, among them the inhibition of the synthesis of pro-inflammatory cytokines, reduction of leucocyte trafficking, and induction of apoptosis of T-lymphocytes” (lines 193-195, page 5, of the revised manuscript).
“TNF-α is a pro-inflammatory cytokine that recruits neutrophils and monocytes to the inflammation areas and activates intracellular signaling in several cells of the immune system” (lines 217-219, page 6, of the revised manuscript).
“B-cells are an essential component of the adaptive immune system, in fact” (line 243, page 7, of the revised manuscript).
“The JAK/signal transduction and activators of transcription (STAT) pathway is responsible for signal transductions triggered by a number of cytokines and growth factors. Thus” (lines 273-275 page 8, of the revised manuscript).
“IL-6 is a very important pro-inflammatory cytokine, involved in the acute phase response and in the differentiation and function of B and T cells” (lines 301-303 page 8, of the revised manuscript).
“Abatacept, is a fusion protein of cytotoxic T lymphocyte- associated antigen-4 that selectively modulates the CD80/CD86:CD28 costimulatory signal required for full T-cell activation” (lines 318-320, page 9, of the revised manuscript).
The critical period in the generation of the humoral response is 2 weeks after vaccination. The interruption of MTX 2 weeks after vaccination is the one that obtains the best result, enhancing the humoral response to the vaccine. Is this good for all Rheumatoid Arthritis patients no matter how severe the symptom it is? How to evaluate?
Answer: The reviewer is right, and brings to light an extremely interesting topic, as we try to reflect in the paper and is currently accepted by most of the authors based on current evidence, the ideal in the case of MTX would be to rest for 2 weeks, however this is only possible if the patient's situation allows it, since if the patient presents the minimum risk of suffering a flare-up, this situation should be prioritized and the drug should not be temporarily suspended.
Reviewer 3 Report
Comments and Suggestions for Authors
This paper addresses a very interesting topic and still not well understood like COVID-19 vaccination schedule in immunosuppressed patients.However, I noticed a not clear organization of this article.
In particular:
1) Introduction should be implemented: the aim of this study should be clearly stated (lines 81-82). Also lines 328-353 seem to belong to Introduction section.
2)Methodology needs to be clarified (type of paper, criteria etc.).
3) Results and Discussion are not specified.
4) Limitation are not mentioned
Therefore, I would suggest a major revision of the structure of this paper.
Author Response
This paper addresses a very interesting topic and still not well understood like COVID-19 vaccination schedule in immunosuppressed patients. However, I noticed a not clear organization of this article.
In particular:
1) Introduction should be implemented: the aim of this study should be clearly stated (lines 81-82).
Answer: We thank the reviewer for the suggestion and we agree to simplify the introduction. In fact, we eliminated the contents of the section called introduction and took the following section as such. Certainly, all the comments in this section do not contribute anything of value to the article. Consequently, we believe this change makes the content of the paper easier to read. (Data erased lines 32-47…pages 1-2)
Also lines 328-353 seem to belong to Introduction section.
Answer: We thank the reviewer for his/her comment, Accordingly, we have erased one paragraph of the lines suggested as the information given in it is already clear in previous paragraphs (lines 349-352, page 10, of the revised manuscript).
2) Methodology needs to be clarified (type of paper, criteria etc.).
Answer: We appreciate the reviewer's comment and are sorry for not being clear enough. This is a review article in which we have tried to include all the information available so far on the studies carried out on the subject in question. In order to subsequently try to generate a series of recommendations, based precisely on all the information reviewed.
3) Results and Discussion are not specified.
Answer: We appreciate the reviewer's clarification in this regard. It is true that we have not titled any section as "discussion", however the section "individualize vaccination program in patients with IMIDs" is the discussion of the work. We have therefore changed the name of the paper to "discussion" (line 333, page 9, of the revised manuscript).Once again, we thank the reviewer for his contribution, as this makes it much clearer and more structured.
4) Limitation are not mentioned.
Answer: Once again, we appreciate the reviewer's comments and agree with them, so we have added a section on limitations at the end of the article, before funding. (line 416, page 13, of the revised manuscript).
Therefore, I would suggest a major revision of the structure of this paper.
Answer: We apologize for certain errors in the structure of the paper and thank the reviewer for his/her comments, which have cotributed to improve the quality of the article.
Round 2
Reviewer 1 Report
Comments and Suggestions for Authors
No comments
Author Response
We would like to thank again the reviewer for all the positive appraisal of our study and his/her comments. Please, find below all the point-by-point replies to the queries requested.
- Authors should summarize several shared characteristics of immunosuppressive drugs and concise the whole manuscript as a mini review. Several information could be demonstrated within the tables.
Answer: We thank the reviewer for his/her suggestion. We have undertaken the proposed changes, summarizing the information common to the different immunomodulatory drugs and reducing the extension of the entire manuscript to give it the format of a mini review.
We have eliminated this information since it is sufficiently clear in this section the management we propose for MTX and it is also commented again in the discussion (Removed data: lines 238-248…page 5).
This is clear in the discussion and throughout the paper (Deleted data: lines 301-306…page 7).
It was not our goal to review so much information on behavior with other vaccines: (Deleted data: lines 323-330…page 9).
It was not our goal to review so much information on behavior with other vaccines: (Removed data: lines 359-369…page 10).
Following the reviewer's advice, we also consider that this statement made in the conclusion of the paper has already been commented on: (Deleted data: lines 437-440…page 12).
We also removed the title "IMPACT OF IMMUNOSUPRESSIVE THERAPY ON COVID 19 VACCINE RESPONSES" as each section perfectly describes the topic to be addressed (Removed data: lines 139-140…page 3).
In addition, we have made small minor changes to make sense of the new changes, so we do not specify point by point but you can see them in the version with change control.
In this process, we have removed some references that we considered were not essential to the work as the editors commented: 3-6;8-10;5,6;27; 28; 49-51; 61-63; 72; 73; 78; 91-94.
- Authors should emphasize primarily the efficacy of vaccines in the pandemic, but not solely on immune responses. In this regard, especially the conclusion as well as the corresponding figure should be revised.
Answer: We thank the reviewer for this comment. Clinical efficacy in terms of reduction of COVID incidence, hospitalizations and deaths is crucial. The information available in this regard is still scarce in patients with IMIDs and we still do not have enough data to allow us to draw robust conclusions about the impact of DMARDs at this level. We have included a mention about it with its references. (Added data: lines 468-520…pages 12-13). In this process, we have added some references:76-80.

Reviewer 3 Report
Comments and Suggestions for Authors
Paper structure was correctly revised. However, "Methods" paragraph is not present and this is essential when you are doing a Review. In particular, you should describe study design, where and why you choose articles you mentioned etc. Therefore, Limitation paragraph should be included in the text in Discussion part.
Author Response
We thank the reviewer for the very pertinent comment. We agree that it is essential to add a "Methods" section( lines 141-155…pages 3-4), which we have incorporated. In the same way, we have added the section “Results” (line 170).
In the same way, and as pointed out by the reviewer, we have eliminated the limitations section to include it in the discussion (Lines section 478-492..page 13).

Round 3
Reviewer 1 Report
Comments and Suggestions for Authors
Dear authors
Your manuscript is now acceptable